# Synthesis, Pharmacokinetic Profile, Anticancer Activity and Toxicity of the New Amides of Betulonic Acid—In Silico and In Vitro Study

**DOI:** 10.3390/ijms25084517

**Published:** 2024-04-20

**Authors:** Ewa Bębenek, Zuzanna Rzepka, Justyna Magdalena Hermanowicz, Elwira Chrobak, Arkadiusz Surażyński, Artur Beberok, Dorota Wrześniok

**Affiliations:** 1Department of Organic Chemistry, Faculty of Pharmaceutical Sciences in Sosnowiec, Medical University of Silesia in Katowice, 4 Jagiellońska, 41-200 Sosnowiec, Poland; ebebenek@sum.edu.pl (E.B.); echrobak@sum.edu.pl (E.C.); 2Department of Pharmaceutical Chemistry, Faculty of Pharmaceutical Sciences in Sosnowiec, Medical University of Silesia in Katowice, 4 Jagiellońska, 41-200 Sosnowiec, Poland; zrzepka@sum.edu.pl (Z.R.); abeberok@sum.edu.pl (A.B.); 3Department of Pharmacodynamics, Medical University of Bialystok, Mickiewicza 2c, 15-222 Bialystok, Poland; justyna.hermanowicz@umb.edu.pl; 4Department of Clinical Pharmacy, Medical University of Bialystok, Mickiewicza 2c, 15-222 Bialystok, Poland; 5Department of Medicinal Chemistry, Medical University of Bialystok, Kilinskiego 1, 15-089 Bialystok, Poland; arkadiusz.surazynski@umb.edu.pl

**Keywords:** betulonic acid, anticancer potential, melanoma, breast cancer

## Abstract

Betulonic acid (B(O)A) is a pentacyclic lupane-type triterpenoid that widely exists in plants. There are scientific reports indicating anticancer activity of B(O)A, as well as the amides and esters of this triterpenoid. In the first step of the study, the synthesis of novel amide derivatives of B(O)A containing an acetylenic moiety was developed. Subsequently, the medium-soluble compounds (EB171 and EB173) and the parent compound, i.e., B(O)A, were investigated for potential cytotoxic activity against breast cancer (MCF-7 and MDA-MB-231) and melanoma (C32, COLO 829 and A375) cell lines, as well as normal human fibroblasts. Screening analysis using the WST-1 test was applied. Moreover, the lipophilicity and ADME parameters of the obtained derivatives were determined using experimental and in silico methods. The toxicity assay using zebrafish embryos and larvae was also performed. The study showed that the compound EB171 exhibited a significant cytotoxic effect on cancer cell lines: MCF-7, A-375 and COLO 829, while it did not affect the survival of normal cells. Moreover, studies on embryos and larvae showed no toxicity of EB171 in an animal model. Compared to EB171, the compound EB173 had a weaker effect on all tested cancer cell lines and produced less desirable effects against normal cells. The results of the WST-1 assay obtained for B(O)A revealed its strong cytotoxic activity on the examined cancer cell lines, but also on normal cells. In conclusion, this article describes new derivatives of betulonic acid—from synthesis to biological properties. The results allowed to indicate a promising direction for the functionalization of B(O)A to obtain derivatives with selective anticancer activity and low toxicity.

## 1. Introduction

Cancer is currently the first or second most common cause of premature death in most countries. The incidence of all cancers combined is projected to double by 2070 compared to 2020 [1]. There is still a lack of effective treatments for many types of cancer, including breast cancer and melanoma, so the search for new compounds with potential selective efficacy against tumor cells seems reasonable. An interesting approach is to synthesize novel derivatives of plant compounds with known anticancer potential in order to obtain optimal compounds both in terms of biological activity and the toxicological and pharmacokinetic profiles. Biologically active compounds of natural origin are characterized by structural diversity, which makes them an important source of obtaining new drugs. About 60% of natural products or their derivatives are used in the treatment of cancer. In this group of compounds, substances of plant origin show significant potential as anticancer or chemopreventive agents [2,3].

Triterpenic acids, secondary metabolites of plants, are found in peels, leaves, stems, roots and barks of various plants species. These compounds have anticancer, antibacterial, antioxidant, anti-inflammatory and hepatoprotective properties without causing toxic effects [4].

Betulonic acid (3-oxo-lup-20(29)-28-oic acid, B(O)A) is a pentacyclic lupane-type triterpenoid isolated from popular fruits in China, such as *Punica granatum* (seed), *Crataegus pinnatifida* (sarcocarp, seed), *Ziziphus montana* (sarcocarp, seed) and *Citrus reticulata* (seed) [5]. This triterpenic acid is also extracted from the bark of *Acacia mellifera*, the roots of *Ficus macrocarpa*, *Ainsliaea acerifolia* and *Toona sinensis*, the stem of *Viscum coloratum* and the leaves of *Lantana camara* L. [3,6,7]. Although betulonic acid can be isolated from many plants, its low content makes extraction from natural sources unprofitable. An alternative method of obtaining B(O)A is the selective oxidation of betulin using Jones reagent (CrO_3_, H_2_SO_4_) [8].

Betulonic acid has an inhibitory effect against lung (A549), ovarian (SK-OV-3), melanoma (SKMEL-2, B16F10 and LOX-IMVI), glioblastoma (XF498), adenocarcinoma (HT15 and PC-3), leukemia (K562), gastric mucinous adenocarcinoma (MGC-803), breast (MCF-7) and prostate (DU-145 and LNCaP) cancer cell lines [3,6,9]. Further studies to elucidate the mechanism of the anticancer effect of B(O)A in MGC-803 cells showed that it induces apoptosis through the mitochondrial signaling pathway, including the expression of p53, Bax and caspases 9 and 3 [3]. The chemical modification of the carboxylic group of betulonic acid at the C17 position to corresponding analogues containing an amide or ester moiety usually leads to an increase in activity. Betulonic acid amides/esters are often more effective inhibitors of cancer cell growth than the parent compound. The conducted studies have shown that these compounds have antiproliferative activity against a range of human cancer lines (MT-4, MOLT-4, CEM, Hep G2, Bel-7402, HeLa, MCF-7, Bcap-37, MGc-803, PC3 and A375; Figure 1) [10,11,12]. Moreover, amide and ester derivatives of betulonic acid have been found to possess antiviral, anti-inflammatory and antibacterial properties (Figure 1) [13,14,15,16,17].

We have previously indicated the anticancer potential of the indole-functionalized derivatives of betulin [18,19]. The strongest effect was obtained for melanoma and breast cancer cells. In this work, we have developed the synthesis of novel amide derivatives of B(O)A. The lipophilicity and the pharmacokinetic profile of the obtained compounds were determined. The compounds were then investigated for potential cytotoxic effects on breast cancer and melanoma cell lines. Human fibroblasts were used for a cytotoxicity assessment to normal cells in vitro. A toxicity assay using zebrafish (*Danio rerio*) embryos and larvae was conducted for the most promising derivative.

We believe that our research will expand the knowledge of the effect of the chemical structure on the pharmacological activity of amide derivatives of betulonic acid. This may contribute to obtaining compounds with selective activity against cancer cells and potential application in oncology.

## 2. Results and Discussion

### 2.1. Chemistry

Betulin, a natural pentacyclic triterpene diol, is isolated as a white substance from birch bark [20]. The efficient synthesis of B(O)A using betulin as a starting material was performed according to the method described by Kim et al. [21]. The C3- and C28-hydroxyl groups of betulin were oxidized with Jones reagent (CrO_3_, H_2_SO_4_, 0 °C) in acetone, obtaining betulonic acid at a 73% yield. The chemical structure of B(O)A was confirmed based on the ^1^H and ^13^C NMR spectra, which were in accordance with analytical data available in the literature [22]. Betulonic acid was reacted with oxalyl chloride (COCl)_2_ in dichloromethane to yield betulonic acid chloride as an intermediate. To synthesize new amide derivatives, acylation of the corresponding acetylenic amines with betulonic acid chloride was performed. The acid chloride was reacted with the appropriate amine in dichloromethane in the presence of triethylamine (Et)_3_N to produce amide derivatives EB170, EB171 and EB173 at a 70–77% yield. The obtained acetylenic amides of B(O)A were purified by column chromatography. The structures of the synthesized compounds were determined by spectroscopic methods, including ^1^H NMR, ^13^C NMR, IR, EI MS, and HRMS (Appendix A). The reagents and conditions applied in the two-step synthesis of new amides are shown in Figure 1.

### 2.2. Lipophilicity and ADME Parameters

Drug design is a long-term, very expensive process with a high risk of failure. Currently, in order to minimize these aspects, quick and efficient in silico methods are widely used, chemical modifications of compounds with known effects are carried out to obtain substances with a more favorable pharmacological profile, and known drugs are tested for other therapeutic directions [23].

The search for new medicinal substances requires detailed knowledge of their physicochemical properties and prediction of their pharmacokinetic profile. Among the molecular physical properties, lipophilicity, expressed as the decimal logarithm of the partition coefficient between the organic and aqueous phases (logP), is a very important parameter determining the fate of a drug in the body [24]. Experimental and in silico methods are used to determine the value of this parameter.

The study of the lipophilicity of the compounds synthesized in this work was carried out experimentally using reversed-phase thin-layer chromatography (RP-TLC). The measurement conditions and description of the procedure are provided in the Appendix A.

In the group of tested compounds, the lowest lipophilicity was determined for the EB170 derivative characterized by a branched alkynyl substituent at the nitrogen atom of the amide group, and it was lower than the value determined for the B(O)A, taken as the reference compound (Table 1). The highest value of the logP_TLC_ parameter determined experimentally was for the derivative EB173 with a cyclic substituent.

In the process of searching for new medicinal substances, in silico methods are increasingly used. Based on five computer programs (iLOGP, XLOGP3, WLOGP, MLOGP and SILICOS-IT) available online, the theoretical lipophilicity value was determined for the tested compounds (Appendix A). The theoretical results obtained, regardless of the computational algorithm, indicated B(O)A as the least lipophilic in the tested group. The profile of changes in the theoretical values of the lipophilicity parameter of compounds was compared with experimental values (Figure 2).

Correlations between the theoretical lipophilicity parameters are presented in Appendix A. The calculated correlation coefficients ranged from 0.680 to 0.998. The highest correlation value was obtained for XLOGP3 and WLOGP (0.998).

Physicochemical parameters constitute the basis for rules formulated to determine the drug similarity of new derivatives. The most frequently used is the Lipiński rule (lipophilicity, MlogP ≤ 4.15, molecular weight, MW ≤ 500, and the number of hydrogen bond acceptors, HA ≤ 10, and donors, HD ≤ 5) [25]. The drug similarity rule according to Ghose takes into account the lipophilicity value, expressed as WLOGP (range from −0.4 to 5.6), molecular weight (range 160 to 480), molecular refraction (RefM, range 40 to 130) and the total number of atoms in the molecule (range 20 to 70) [26]. Drug-likeness according to Veber depends on the number of rotational bonds (RB ≤ 10) and the topological value of the polar surface of the molecule (TPSA, below 140 Å^2^) [27]. The TPSA value is a descriptor determining the ability of molecules to penetrate biological membranes, and thus characterizing the bioavailability of a compound. The TPSA value defined by Veber, below 140 Å^2^, determines good absorption of substances administered orally [27]. All tested compounds only meet Veber’s drug similarity rule. The values of molecular weight and the lipophilicity parameter are the main reason for exceeding the remaining rules (Table 2).

The initial assessment of a chemical compound as a drug candidate also includes the prediction of the ADMET (absorption, distribution, metabolism, excretion and toxicity) profile using in silico methods. Key parameters related to absorption and distribution are the permeability of Caco-2, skin and the blood–brain barrier (BBB). Substances with high Caco-2 permeability (logPapp in 10^−6^ cm/s > 0.9) are considered to be easily absorbed in the small intestine. LogPapp values of tested derivatives were in the range of 1.270–1.340 for compounds with good permeability. Skin permeability (logKp) determines the ability of compound molecules to penetrate the skin. The logKp values of the tested compounds determined an in silico range from −2.771 cm/h to −2.729 cm/h, which indicates their good permeability through the skin. LogKp values < −2.5 cm/h characterize substances with good skin permeability, which is important in the case of local use of drugs [28,29].

The BBB is a specific protection of the brain that prevents the penetration of polar molecules from the outside. Compounds with high lipophilicity penetrate the BBB by diffusion, while less lipophilic substances penetrate through active transport using appropriate carriers. Determination of the blood–brain partition coefficient (logBB) is an important factor taken into account in the case of substances designed as potential drugs acting on the central nervous system. It has been shown that molecules with a logBB value > 0.3 easily cross the blood–brain barrier, while molecules with a logBB value < −1 are poorly distributed to the brain [30].

B(O)A and its amide derivatives are characterized by moderate permeability through the BBB. The permeability into the central nervous system (CNS) can be determined based on the blood–brain permeability area product (logPS). Compounds may enter the central nervous system when the predicted logPS value is > −2 [31,32,33]. Betulonic acid and derivatives EB170 and EB171 have the ability to penetrate the CNS barrier, as indicated by logPS values of −1.160, −1.662 and −1.761.

The pkCSM platform predicts properties related to the safety of using tested compounds as medicinal substances, e.g., cytotoxicity toward HepG2 cells, mutagenicity (AMES assay) and cardiotoxicity (hERG block). Predictions obtained from the pkCSM program for B(O)A and its amide derivatives indicate the lack of mutagenic, cardiotoxic, hepatotoxic and skin-sensitizing effects (Table 3).

Appendix A shows the values of correlation coefficients between molecular descriptors determined using in silico methods and the experimental value of the lipophilicity parameter. A high correlation between lipophilicity and logPapp, logBB and logPS parameters has been shown.

### 2.3. Screening Analysis for Anticancer Potential

To evaluate the anticancer (anti-melanoma and anti-breast-cancer) activity of the newly synthesized amides of betulonic acid, the WST-1 assay was performed in two-dimensional culture of various human cancer cell lines: MC7-7 (breast cancer with estrogen, progesterone and glucocorticoid receptors), MDA-MB-231 (triple-negative breast cancer), C32 and A375 (amelanotic melanoma cells) and COLO 829 (melanotic melanoma cells). Normal human fibroblasts (HDFs) were also studied to assess the selectivity of the compounds’ cytotoxic action against cancer cells. Due to the poor solubility of EB170 in culture media, this compound was not used in our cytotoxicity studies. The results obtained for EB171 and EB173 are shown in Figure 3. Moreover, an experimental panel was also conducted for B(O)A in order to compare the effects produced by the parent compound.

Based on the results, EB171 was found to have a stronger effect on MCF-7 cells than on MDA-MB-231 cells—the compound at a concentration of 20 µM reduced the viability of MCF-7 cells by about 40% compared to the control, while on MDA-MB-231 cells a similar effect requires a concentration of 100 µM (Figure 3). For the tested melanoma cell lines, the strongest cytotoxic effect was observed for the A375 cells (IC_50_ = 17 µM), a less pronounced effect for the COLO 829 cell line (IC_50_ = 35 µM) and the most resistant melanoma cell line was C32 (Table 4). It is noteworthy that for the A375 line, a concentration of 20 µM reduced the survival rate to ca. 30% of the control. As presented in Figure 3 and Table 4, the compound EB171 did not induce cytotoxicity on HDFs over the entire concentration range, indicating its selectivity toward tumor cells.

Compared to EB171, compound EB173 had a weaker effect on all cancer cell lines tested (Figure 3 and Table 4). For the latter derivative, we observed the most significant reduction in survival rate for melanoma cell lines COLO 829 (IC_50_ = 131 µM) and A375 (IC_50_ = 160 µM). In addition, EB173 showed less desirable effects against normal cells—after incubation with EB173 at a concentration of 20–200 µM, we observed a significant reduction in HDFs’ viability by 14–45%, compared to the control.

There are previous studies aimed at evaluating the anticancer activity of various derivatives of betulonic acid with modifications at the C17. The largest group of such compounds are ester derivatives [10,11,12].

This article presents data on the anti-melanoma and anti-breast-cancer potential of the betulonic acid amides containing an acetylenic moiety. The obtained results indicated that the transformation of the carboxyl group into an amide function containing a carbon–carbon triple bond may lead to an increase in activity against cancer cells. The structure of the substituent in the amide group had a significant impact on the activity. The compound EB173, containing a large cyclic substituent (1-ethynylcyclohexyl), showed lower activity against the tested lines compared to betulonic acid. Replacing the 1-ethynylcyclohexyl substituent with an N-methylpropargyl group in the EB171 compound resulted in a significant increase in activity against MCF-7 breast cancer cells, as well as A375 and COLO 829 melanoma cells.

Our cytotoxicity studies of B(O)A showed that the compound had a strong effect on reducing the survival of all cell lines tested (Figure 3). For the A375 line, the IC_50_ of betulonic acid was only 7 µM (Table 4). Nevertheless, B(O)A had strong cytotoxic effects on normal cells (IC_50_ for HDFs: 14 µM), which may suggest a toxicity of this compound to normal tissues. Therefore, betulonic acid has not been subjected to further in vivo testing. The cytotoxicity of B(O)A was previously determined by other authors. Tsepaeva et al. [17] published MTT assay data showing that the IC_50_ of betulonic acid on human normal dermal fibroblasts was 12.5 µM. This result is similar to that obtained in our study.

The determined lipophilicity of betulonic acid, EB171 and EB173 showed a correlation with the antitumor activity (IC_50_) determined against C32, Colo829, A375 and MCF-7 cells (Appendix A).

### 2.4. The Effects of EB171 on Zebrafish Embryo/Larvae Development

Taking into account the results of the in vitro cytotoxicity study, it was considered that the EB171 derivative was the most promising, showing the greatest anticancer potential and no effect on normal cells. Therefore, this compound was selected for further studies to test toxicity in a zebrafish model.

#### 2.4.1. Embryos of 0–2 hpf

The survival and early embryonic development were examined at 4, 8, 12, 24, 48, 72 and 96 h after exposure to EB171. There were no significant differences between untreated embryos and embryos incubated with 1% DMSO. Mortality or developmental malformations in untreated embryos was not recorded at any time of the observation. Figure 4A shows that at 96 h post-exposure, the recorded survival was 83% and 80%, respectively, in the group of embryos exposed to EB171 at 50 and 100 μM. The most prominent effects on the embryo survival and development were present at the beginning of treatment in these studied groups. Embryo phenotypic features were analyzed at 24 hpf and 96 hpf. As shown in Figure 4C, malformations of zebrafish embryos exposed to EB171 at the concentration of 100 μM were observed in 20% (*** *p* < 0.001 vs. CON), whereas at the concentration of 50 μM, malformations were at the control level.

The embryo hatching rate, determined by counting of zebrafish larvae outside the eggshell, was observed at 48 and 72 hpf, as hatching normally occurs during this period. The hatching rate of untreated embryos was approximately 11% at 48 hpf and 100% at 72 hpf. About 3% less embryos treated with 50 μM of EB171 hatched at 48 hpf, whereas EB171 at the concentration of 100 μM caused the hatching of 25% of the embryos (Figure 5A).

Exposure to EB171 affected the cardiac function of zebrafish embryos, reflected as HR (heart rate), and the concentration of 50 μM significantly decreased the HR compared to the control (Figure 5C). We noticed that embryo development was slowed when treated with 50 μM (* *p* < 0.05) and 100 μM (** *p* < 0.01) of EB171, reflected as a reduction in total body length (96 hpf; Figure 5B). According to Figure 5D, body pigmentation after exposure to higher concentrations, such as 50 μM and 100 μM, was significantly lower compared to untreated embryos (*** *p* < 0.001). The results indicated that EB171 affects pigmentation, which could be investigated in a future study.

#### 2.4.2. Larvae of 72 hpf

Then, we performed a toxicity study for EB171 in 72 hpf larvae. At all tested concentrations, it did not cause significant larvae death throughout the assay (Figure 4B). Moreover, no significant malformations or developmental abnormalities were observed during the evaluation until the end point (Figure 4D). The obtained results indicated the lack of toxicity of the tested compound in an animal model.

## 3. Materials and Methods

### 3.1. Synthesis

#### 3.1.1. Materials and Methods Included in the Appendix A

Betulonic acid was synthesized based on a previously described method. Data on the melting point and ^1^H and ^13^C NMR spectra of betulonic acid were consistent with information from the literature [22].

#### 3.1.2. Synthesis of Amide Derivatives of Betulonic Acid EB170, EB171 and EB173

B(O)A at 0.25 g (0.55 mmol) was dissolved in dry dichloromethane (8 mL), and oxalyl chloride at 0.069 mL (0.1 g, 0.83 mmol) was added. Then, the reaction mixture was stirred at room temperature for 24 h. Dichloromethane was then distilled off under reduced pressure. The resulting solid was treated twice with the same amount of cyclohexane (6 mL), which was removed after each addition. In this way, 0.20 g of crude betulonic acid chloride was obtained. The obtained chloride was dissolved in dry dichloromethane (5.8 mL). The solution of the appropriate amine: 2-methyl-3-butyn-2-amine or *N*-methylpropargylamine or 1-ethynylcyclohexylamine (0.61 mmol) in dichloromethane (1.7 mL), was added dropwise. Finally, triethylamine at 0.16 mL (0.12 g, 1.21 mmol) was added. The reaction mixture was stirred at room temperature for 24 h. The mixture was then diluted with 12 mL of dichloromethane. The organic layer was washed twice with water, dried over anhydrous sodium sulfate and concentrated under reduced pressure. The crude reaction products were purified by column chromatography (SiO_2_, eluent: chloroform/ethanol, 40:1, *v*/*v*).

*N*-[3-oxolup-20(29)-en-28-oyl]2-methyl-3-butyn-2-amine **EB170**Yield 72%; mp 231–233 °C; R_f_ 0.61 (chloroform/ethanol, 40:1, *v*/*v*).^1^H NMR (600 MHz, CDCl_3_) δ: 0.93 (s, 3H, CH_3_), 0.98 (s, 3H, CH_3_), 1.00 (s, 3H, CH_3_), 1.03 (s, 3H, CH_3_), 1.08 (s, 3H, CH_3_), 1.69 (2 x s, 6H, C(CH_3_)_2_), 1.71 (s, 3H, CH_3_), 0.95–2.03 (m, 20H, CH, CH_2_, from the basic lupane system), 2.31 (s, 1H, C≡CH), 2.41 (m, 1H, CH, from the basic lupane system), 2.49 (m, 1H, CH, from the basic lupane system), 2.62 (m, 1H, from the basic lupane system), 3.19 (m, 1H, H-19), 4.60 (m, 1H, H-29), 4.74 (m, 1H, H-29), 5.60 (s, 1H, N-H); ^13^C NMR (150 MHz, CDCl_3_) δ: 13.5, 14.8, 14.9, 18.6, 18.6, 20.0, 20.4, 24.6, 25.6, 27.8, 28.0, 28.4, 29.8, 32.7, 33.0, 33.1, 35.9, 36.5, 37.2, 38.6, 39.8, 41.5, 45.4, 46.1, 46.3, 49.0, 49.1, 54.0, 54.7, 67.7, 86.5, 108.2, 150.0, 174.2, 217.2; IR (ν max cm^−1^, KBr): 1447, 1661, 1708, 2938, 3244, 3436; EI MS (70 eV) *m*/*z* (rel. intensity): 519 (M+, 100), 203 (46), 189 (35); HRMS (APCI) m/z (neg): 518.4005; C_35_H_52_NO_2_ (Calculated 518.3998).*N*-[3-oxolup-20(29)-en-28-oyl]methylpropargilamine **EB171**Yield 77%; mp 179–181 °C; R_f_ 0.71 (chloroform/ethanol, 40:1, *v*/*v*).^1^H NMR (600 MHz, CDCl_3_) δ: 0.94 (s, 3H, CH_3_), 0.98 (s, 3H, CH_3_), 0.99 (s, 3H, CH_3_), 1.03 (s, 3H, CH_3_), 1.07 (s, 3H, CH_3_), 1.70 (s, 3H, CH_3_), 0.95–1.94 (m, 18H, CH, CH_2_, from the basic lupane system), 2.05 (m, 1H, C≡CH), 2.25 (m, 2H, CH_2,_ from the basic lupane system), 2.41 (m, 1H, CH, from the basic lupane system), 2.50 (m, 1H, CH, from the basic lupane system), 2.90 (m, 1H, CH, from the basic lupane system), 3.00 (m, 1H, H-19), 3.11 (br s, 3H, NCH_3_), 4.17 (m, 2H, NCH_2_), 4.59 (m, 1H, H-29), 4.74 (m, 1H, H-29); ^13^C NMR (150 MHz, CDCl_3_) δ: 13.6, 14.9, 15.0, 18.6, 18.7, 20.0, 20.6, 24.6, 25.6, 28.8, 30.3, 31.0, 32.7, 33.2, 34.8, 35.9, 36.0, 38.7, 39.6, 40.9, 44.6, 46.3, 49.2, 51.6, 53.7, 54.1, 108.2, 150.3, 173.5, 217.3; IR (ν max cm^−1^, KBr): 1460, 1642, 1703, 2118, 2960, 3247; EI MS (70 eV) *m*/*z* (rel. intensity): 505 (M+, 72), 409 (100), 203 (21), 189 (72); HRMS (APCI) *m*/*z* (neg): 504.3850; C_34_H_50_NO_2_ (Calculated 504.3842).*N*-[3-oxolup-20(29)-en-28-oyl]1-ethynylcycloheksylamine **EB173**Yield 70%; mp 194–196 °C; R_f_ 0.68 (chloroform/ethanol, 40:1, *v*/*v*).^1^H NMR (600 MHz, CDCl_3_) δ: 0.93 (s, 3H, CH_3_), 0.98 (s, 3H, CH_3_), 1.01 (s, 3H, CH_3_), 1.03 (s, 3H, CH_3_), 1.08 (s, 3H, CH_3_), 1.39–1.61 (m, 10H, CH, CH_2_, from cyclohexyl ring), 1.69 (s, 3H, CH_3_), 0.95–2.16 (m, 19H, CH, CH_2_, from the basic lupane system), 2.37 (s, 1H, C≡CH), 2.41 (m 1H, CH, from the basic lupane system), 2.50 (m, 1H, CH, from the basic lupane system), 2.64 (m, 1H, CH, from the basic lupane system), 3.19 (m, 1H, H-19), 4.59 (m, 1H, H-29), 4.74 (m, 1H, H-29), 5.50 (s, 1H, NH); ^13^C NMR (150 MHz, CDCl_3_) δ: 13.5, 14.9, 14.9, 18.6, 20.0, 20.4, 21.4, 24.3, 24.6, 25.6, 28.4, 29.8, 32.7, 33.0, 33.1, 35.6, 35.9, 36.4, 37.4, 38.6, 39.8, 41.5, 45.4, 46.3, 49.0, 49.1, 50.2, 54.0, 54.7, 54.8, 69.7, 85.0, 108.2, 150.1, 174.0, 217.2; IR (ν max cm^−1^, KBr): 1458, 1669, 1701, 2937, 3309; 3391; EI MS (70 eV) *m*/*z* (rel. intensity): 559 (M+, 90), 531 (100), 505 (42), 189 (15); HRMS (APCI) *m*/*z* (neg): 558.4310; C_38_H_55_NO_2_ (Calculated 558.4311).

#### 3.1.3. Lipophilicity

The methodology for experimental determination of lipophilicity (logP_TLC_ parameter) using reversed-phase thin-layer chromatography is presented in the Appendix A.

#### 3.1.4. In Silico Analysis

In order to perform analysis using in silico methods, first, the chemical structures of the tested molecules were transformed into SMILES (simplified molecular-input line-entry system) codes using the Chem Draw program (Perkin Elmer Informatics, Waltham, MA, USA).

The calculations of the theoretical values of the logP parameter were performed using the online SwissADME software (http://www.swissadme.ch (accessed on 21 November 2023)). The parameters necessary to assess drug similarity and conduct a preliminary characterization of the ADMET profile of the tested compounds were calculated using the pkCSM-Biosig Lab—University of Melbourne server (http://biosig.unimelb.edu.au (accessed on 21 November 2023)).

### 3.2. Biological Activity

#### 3.2.1. Cell Culture

Human breast cancer MC7-7 and MDA-MB-231 cells, as well as human melanoma C32 cells and COLO 829 cells, were obtained from the American Type Culture Collection (ATCC). The base medium for MCF-7, MDA-MB-231, C32 and A375 was Dulbecco’s modified Eagle’s medium (DMEM), from Cell Applications, San Diego, CA, USA. The base medium for COLO 829 was Roswell Park Memorial Institute (RPMI) 1640 (Cell Applications). To make the complete growth media, the following components were added: fetal bovine serum (FBS; Thermo Fisher Scientific, Waltham, MA, USA) to a final concentration of 10%, penicillin G (final concentration: 100 U/mL), neomycin (final concentration: 10 µg/mL) and amphotericin B (final concentration: 0.25 µg/mL). Normal dermal human fibroblasts were obtained from Sigma-Aldrich and cultured in Fibroblast Growth Medium (Sigma Aldrich, Saint Louis, MO, USA). All cells were maintained at 37 °C in humidified incubators with 5% carbon dioxide. The cells were passaged when there was ca. 80% confluence.

#### 3.2.2. Cell-Based Cytotoxicity Assay

The Cell Proliferation Reagent WST-1 (from Roche, Mannheim, Germany) was used to assess the viability of the cell lines in the presence of the tested compounds. The principle of this colorimetric assay is based on the conversion of the tetrazolium salt to formazan (maximum absorption at 440 nm) by mitochondrial dehydrogenases in living cells. In brief, cells were seeded into 96-well plates (2500 cells/well) and incubated for 24 h. The medium was then replaced with EB171, EB173 or betulonic acid solutions (5, 10, 20, 50, 100 and 200 µM) at a volume of 100 µL/well and incubated for 72 h. Dimethyl sulfoxide (DMSO; Sigma-Aldrich) was used as a drug solvent. The final concentration of DMSO did not exceed 1%, which was assessed by the WST-1 assay as non-toxic for all tested cell lines. WST-1 reagent was added (10 µL per well) 2 h before the end of the incubation period. Absorbance was read at 440 and 650 nm using the microplate reader Infinite 200 Pro (TECAN, Männedorf, Switzerland). The mean of the control wells was taken as 100%. Dose–response curves and the half-maximal inhibitory concentration (IC_50_) were calculated using GraphPad Prism 8 software.

#### 3.2.3. Zebrafish Husbandry

To ensure the well-being of the zebrafish embryos, they were carefully housed in a specially controlled environment. This included maintaining a consistent temperature of 28.0 ± 1.0 °C and providing them with a light/dark cycle that is in line with the guidelines of the Research Animals Department of the esteemed RSPCA (Royal Society for the Prevention of Cruelty to Animals). According to EU Directive 2010/63/EU, the earliest life stages of zebrafish (embryo and eleutheroembryo cultures) are regarded as equivalent to an in vitro cell culture; therefore, they do not fall into the regulatory framework dealing with animal experiments. In our studies, we used zebrafish embryos and larvae younger than 120 hpf (hours post-fertilization); hence, ethical approval was not required. Zebrafish embryos were acquired through the process of mating adult zebrafish. They were then maintained and raised in line with the methods outlined in previous studies [34,35].

#### 3.2.4. Zebrafish Toxicity Assay

The FET (Fish Embryo Toxicity) test was conducted with some modifications [36] (Figure 6). New fertilized wildtype (WT) zebrafish embryos (0–2 hpf) exhibiting normal development or 72 hpf larvae were transferred to 6-well plates filled with a standard E3 medium and solutions of B171 (50 and 100 μM). Dimethyl sulfoxide (DMSO; Sigma-Aldrich) was used as a drug solvent. The final concentration of DMSO in the wells did not exceed the damaging concentration of above 1%. The control embryos were incubated in an embryo medium in the presence of 1% DMSO. The embryos were inspected under a stereomicroscope equipped with a camera at 4, 8, 12, 24, 48, 72 and 96 h of treatment. The experiments were carried out in triplicate and twenty embryos were used for each group. Every 24 h, up to four apical observations were recorded as indicators of lethality: coagulation of fertilized eggs, lack of somite formation, lack of detachment of the tail-bud from the yolk sac and lack of heartbeat. Pigmentation after 48 h and hatching rate after 48, 72 and 96 h were also observed. Additional developmental alterations (heart rate and total body length) and embryo malformations, such as pericardial edema, yolk sac edema, tail curvature, somite formation and scoliosis, were recorded at 96 h.

In the toxicity test with 72 hpf larvae, 3 replicates were performed. For each replicate, 20 objects were used in each concentration and 20 larvae were used as a control (1% DMSO). The larvae were monitored for 24 and 48 h after the treatment. The survival rate and morphological deformities were examined and documented using a stereomicroscope equipped with a camera.

After completing the observations, all remaining embryos/larvae were euthanized using a buffered tricaine methane-sulphonate solution, as per the OECD Test Guideline 236 (Organization for Economic Co-operation and Development 2013).

### 3.3. Statistical Analysis

Shapiro–Wilk’s W test of normality was used for data distribution analysis. The normally distributed data were analyzed using a one-way analysis of variance (ANOVA). Dunnett’s post hoc test was used in the case of the cell-based cytotoxicity assay. For the zebrafish toxicity assay, the relationships between the two variables were analyzed using the Fisher independence test. The statistical analysis was conducted using the GraphPad Prism 9.4 software (Version 9.4.1, GraphPad Software, Boston, MA, USA). The differences were deemed statistically significant when *p* < 0.05.

## 4. Conclusions

This article described the new amides of betulonic acid—from the synthesis and pharmacokinetic profile to the anticancer potential and toxicity. In conclusion, we have demonstrated that the derivative EB171, N-[3-oxolup-20(29)-en-28-oyl]methylpropargilamine, exhibited a significant cytotoxic effect on cancer cell lines: MCF-7, A-375 and COLO 829, while it did not affect the survival of normal fibroblasts, unlike betulonic acid. Moreover, experiments on the zebrafish model indicated no potential toxicity of EB171. The compound EB173, i.e., N-[3-oxolup-20(29)-en-28-oyl]1-ethynylcycloheksylamine, has been shown to be much less selective for anticancer activity. The obtained results allowed to indicate a promising direction of modification of the structure of betulonic acid in order to obtain derivatives with selective anticancer activity and low toxicity. In subsequent studies of amide derivatives of betulonic acid, it would be necessary to analyze the influence of differences in the structure of the amide moiety on the activity of the obtained derivatives. An interesting issue is also the comparison of the action of amide derivatives of betulonic acid and their analogues reduced at the C3 position (derivatives of betulonic acid).

## Data Availability

The raw data supporting the conclusions of this article will be made available by the authors upon request.

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
