# Peer review of "Synthesis, Pharmacokinetic Profile, Anticancer Activity and Toxicity of the New Amides of Betulonic Acid—In Silico and In Vitro Study"

_ijms, 2024, doi:10.3390/ijms25084517_

Round 1
Reviewer 1 Report
Comments and Suggestions for Authors
The manuscript entitled “New amides of betulonic acid – synthesis and anticancer potential” by WrzeÅ›niok et al. (Manuscript Id:2918594) describes the synthesis of new betulonic acid-based derivatives. The authors also described the in silico druggable properties and pharmacological assays of betulonic acid-based amides. However, the following queries need to be addressed before making the decision.
1. Integration of NMR spectra needs attention. The spectral information does not match with the 1H spectral data provided in the manuscript.
2. Cancer introduction in abstract can be deleted and provide information only about the current study.
3. All the scientific names should be in italics. Typographical (Zuco at al line 261 etc) and formatting errors should be corrected in the entire manuscript.
4. Legends for tables with the details of abbreviated forms should be included.
5. Repetition statements can be deleted. Description about BBB in lines 186 and 194 provides the same information.
6. WST-1 assay, provides full form and principle of the assay.
7. Authors tested 5-200 mM concentration in different cell lines, how did authors provide IC50 of 382, 389 etc., for EB173 in some cell lines?
8. Line 261, What did the authors try to explain by reporting Zuco et al. data?
9. Reference formatting should be uniform. E.g., Reference 19 etc.
Comments on the Quality of English Language
Language polishing suggested
Author Response
We thank the Reviewer for constructive comments that were helpful in improving our manuscript.Below is our detailed answer to individual comments.
1. Answer:
We would like to thank the reviewer for his thorough analysis of the manuscript and supplementary materials, which allowed us to detect the error we made. The synthesized compounds were purified several times by column chromatography until pure compounds were obtained that could be submitted for biological research. At each stage of purification, we performed NMR spectra. We have mistakenly included spectra from an earlier stage of purification in the supplement.
2. Answer:
In accordance with the Reviewer's comment, we have removed the first two sentences of the abstract regarding cancer issues.
3. Answer:
Species names are written in italics. Typographical errors and formatting irregularities in the manuscript have been corrected.
4. Answer:
As suggested, explanations of the parameter abbreviations used, their explanation and the units in which they are expressed have been added below Tables 2 and 3.
5. Answer:
The repetition has been removed.
6. Answer:
We have added a description of the principle and procedure of the WST-1 assay (Materials and methods: 3.2.2. Cell-based cytotoxicity assay).
7. Answer:
IC50 values above 200 µM were determined by extrapolation in the statistical program. Nevertheless, we agree with the Reviewer that the parameter so determined may not be appropriate. Therefore, in revised Table 4, instead of providing extrapolated values, we have included "above the tested range”.
8. Answer: In this part of the discussion, we wanted to confirm our result regarding the cytotoxic effect of betulonic acid on normal skin fibroblasts based on other literature reports. Unfortunately, the cited reference referred to betulinic acid and not betulonic acid.
The error has been corrected and the reference has been changed to the correct one (Tsepaeva 2019).
9. Answer:
The formatting of references has been standardized and adapted to MDPI requirements.
Reviewer 2 Report
Comments and Suggestions for Authors
The entitled manuscript “New amides of betulonic acid-synthesis and anticancer potential” by Bebenek et al involved the synthesis of novel organic molecules containing the main core of the betulonic acid. Also, the authors performed several assays to determine the potential action as anticancer agents. Suggestion: please clarify the title because the phrase “anticancer potential” is not so specific for this interesting work.
The introduction of the manuscript offers a comprehensive summary of the state of cancer treatment, highlighting the necessity for effective therapeutic due to the increasing incidence of cancer. The authors have highlighted the implication of utilize novel compounds derived from natural sources for their potential anticancer properties. The rationale behind the reported synthesis of these molecules is well-justified considering its recognized effectiveness against several cancer cell lines. Additionally, the authors have properly cited pertinent literature to support the introduction.
Suggestion: the authors could clearly state the unique contributions of their work and how it advances the field of potential cancer therapeutics.
The methodology includes the synthesis (knowing synthetic routes) and evaluation of betulonic acid derivatives for anticancer potential and biological assays regarding the potential action against breast cancer and melanoma cell lines. Additionally, they used in vivo assays to demonstrate the toxicity of these novel compounds. The methodology is well-designed and suitable for the main goal of the present research.
Suggestion: including mechanistic studies to explain the results will improve the manuscript and research.
The conclusion summarizes the findings, focusing on the synthesis, pharmacokinetics, anticancer potential, and toxicity of novel derivatives of betulonic acid. The authors highlight the cytotoxic effect of the EB171 derivative on multiple cancer cell lines, demonstrating a potential therapeutic advantage over betulonic acid. Additionally, the lack of toxicity observed in zebrafish assays supports the safety profile of EB171.
Suggestion: while the conclusion mentions the potential for further structural modifications to improve selectivity and reduce toxicity, including specific recommendations for future research would enhance precision for subsequent studies in the field.
Please check typo mistakes throughout the manuscript, some examples as it follows:
- N-methylpro-331 pargylamine: N should be in italics.
- C34H50NO2: the quantity of atoms in subscript
- The results should be expressed utilizing the standard deviation (Table 4)
Author Response
We thank the Reviewer for constructive comments that were helpful in improving our manuscript. As suggested by the Reviewer, we have changed the title of the manuscript (the new title: Synthesis, pharmacokinetic profile, anticancer activity and toxicity of the new amides of betulonic acid – in silico and in vitro study). Below is our detailed answer to individual comments.
Suggestion: the authors could clearly state the unique contributions of their work and how it advances the field of potential cancer therapeutics.
Answer:
We have added the following sentence at the end of the Introduction:
We believe that our research will expand the knowledge of the effect of chemical structure on the pharmacological activity of amide derivatives of betulonic acid. This may contribute to obtaining compounds with selective activity against cancer cells and potential application in oncology.
Suggestion: including mechanistic studies to explain the results will improve the manuscript and research.
Answer:
We agree with the reviewer that determining the mechanism of anticancer activity would increase the value of the obtained results. Research in this direction has been planned as an important element of the next publication, the topic of which will be a comparison of the activity and mechanisms of action of amide derivatives of betulonic and betulinic acid containing an acetylenic moiety.
Suggestion: while the conclusion mentions the potential for further structural modifications to improve selectivity and reduce toxicity, including specific recommendations for future research would enhance precision for subsequent studies in the field.
Answer:
As part of the proofreading of the manuscript, a fragment describing the dependence of the activity of the obtained compounds on their structure was added (third paragraph below Table 3). In subsequent studies of amide derivatives of betulonic acid, it would be necessary to analyze the influence of differences in the structure of the amide moiety on the activity of the obtained derivatives. An interesting issue is also the comparison of the action of amide derivatives of betulonic acid and their analogues reduced at the C3 position (derivatives of betulinic acid). An appropriate explanation has been added to the conclusion.
Comment:
Please check typo mistakes throughout the manuscript, some examples as it follows:
- N-methylpropargylamine: N should be in italics.
- C34H50NO2: the quantity of atoms in subscript
- The results should be expressed utilizing the standard deviation (Table 4)
Answer:
Typographical errors have been corrected.
As suggested by the Reviewer, we have added standard deviations to the results in Table 4.
Reviewer 3 Report
Comments and Suggestions for Authors
Dorota et al. submitted the manuscript entitled: New amides of betulonic acid – synthesis and anticancer potential, in which they designed and synthesized 3 betulonic acid derivative as well as tested the cytotoxic activity against different cancer cell lines. Although this topic may be of interest to the potential readers of IJMS, I have one major concern on solubility of these compounds. Please check if the tested compounds can reach the highest concentration in this work (200 uM), otherwise the IC50 values are not reliable.
Some other suggestions:
1. Figure 2: why did the authors use line chart? All the clogp data should be independent and have no relation with other compounds.
2. Table 2: Is Papp data a simulation or experimental data? If it is a simulation, please label it out in table legend. If it is experimental data, please provide method description.
3. Figure 3: please label out each cell line used in figure 3. The current version is confusing.
Author Response
We thank the Reviewer for constructive comments that were helpful in improving our manuscript.
Below is our detailed answer to individual comments.
Comments and Suggestions for Authors
Dorota et al. submitted the manuscript entitled: New amides of betulonic acid – synthesis and anticancer potential, in which they designed and synthesized 3 betulonic acid derivatives as well as tested the cytotoxic activity against different cancer cell lines. Although this topic may be of interest to the potential readers of IJMS, I have one major concern on solubility of these compounds. Please check if the tested compounds can reach the highest concentration in this work (200 uM), otherwise the IC50 values are not reliable.
Answer:
We obtained solutions of the tested compounds (EB171, EB173, and betulonic acid) at a concentration of 200 µM. The compound EB170 is the only compound among the derivatives obtained that is difficult to solubilize and it was not possible to obtain a 200 µM solution. Therefore, we described its synthesis and performed in silico analysis, but we did not take it for cell culture studies.
Suggestion:
Figure 2: why did the authors use line chart? All the clogp data should be independent and have no relation with other compounds.
Answer:
Detailed logP values are included in supplementary materials Table S1 and S2. The main manuscript contains a profile of changes in the theoretical and experimental values of the lipophilicity parameter. We would like to leave this version of the graph because it allows us to observe changes in the logP value depending on the computational program used and compare them with the experimental results.
Suggestion:
Table 2: Is Papp data a simulation or experimental data? If it is a simulation, please label it out in table legend. If it is experimental data, please provide method description.
Answer:
Papp's data are simulation data as indicated in the title of Table 2.
Suggestion:
Figure 3: please label out each cell line used in figure 3. The current version is confusing.
Answer:
According to the Reviewer suggestion, we have labeled each cell line above the graphs.
Round 2
Reviewer 1 Report
Comments and Suggestions for Authors
1. The spectral data doesn’t match with the spectra, for eg, EB170, there are 4 peaks integrated in 1H NMR between 2-3 ppm. But the spectral data accounts for only one proton at 2.31 ppm. What about the others? Are they extra peaks or did authors forget to include those in spectral data?
2. Authors need to properly report the spectral data, for eg, EB171, spectrum has a peak accounting for 6 protons and that is nowhere mentioned in the spectral data. This reviewer recommends integrating and reporting NMR spectra and data clearly.
3. In EB173, if authors want to report 1.39-1.61 as multiplet accounting for 10 protons, the integration of the spectra should also resemble the same.
Comments on the Quality of English LanguageNone
Author Response
Answers to the Reviewer 1
We would like to thank the Reviewer for his comments. New changes in the text are marked in green.
Answer:
Publications containing descriptions of the spectra of betulin and betulonic acid derivatives specify the chemical shifts of protons found in the newly introduced substituents and possibly the shifts of the protons closest to them and the protons in the reactive positions (in this case C30, C29) that have not been modified. Due to the large fragment of the molecule that remains unchanged, signals are not assigned for all the protons of the basic lupane system (CH, CH2). In the case of the spectrum of the compound EB173, the signals of the protons of the cyclohexyl ring appear in the range of shifts for the protons of the basic lupane system.
Description of the spectra of compounds EB170, EB171 and EB173 have been supplemented with information on proton signals (CH, CH2) of the basic lupane system.
Reviewer 3 Report
Comments and Suggestions for Authors
The authors have addressed all the issues.
The authors are suggested to attach the solubility data in SI.
Author Response
Answers to the Reviewer 3
We would like to thank the Reviewer for his comments. The Reviewer's suggestion is valuable. Unfortunately, we do not have solubility data in SI. We hope the Reviewer will not find it necessary to add this information to the manuscript.
Round 3
Reviewer 1 Report
Comments and Suggestions for Authors
Spectral data and spectra are still not matching. Authors should look into it.
Comments on the Quality of English LanguageNone
Author Response
Answers to the Reviewer 1
We would like to thank the Reviewer for his comments. New changes in the text are marked in blue.
Comments:
- The spectral data doesn’t match with the spectra, for eg, EB170, there are 4 peaks integrated in 1H NMR between 2-3 ppm. But the spectral data accounts for only one proton at 2.31 ppm. What about the others? Are they extra peaks or did authors forget to include those in spectral data?
- Authors need to properly report the spectral data, for eg, EB171, spectrum has a peak accounting for 6 protons and that is nowhere mentioned in the spectral data. This reviewer recommends integrating and reporting NMR spectra and data clearly.
3.In EB173, if authors want to report 1.39-1.61 as multiplet accounting for 10 protons, the integration of the spectra should also resemble the same.
Answer:
- Spectral data for EB170 has been corrected. The description of the 1H NMR spectrum includes peaks visible in the range of 2-3 ppm.
- In the EB171 spectrum, the integration of the double peak in the range of 0.98-0.99 ppm has been improved (previously 6 protons). The integration corresponds to two methyl groups (2 x CH3) present in the basic system of lupane.
- In the EB173 spectrum, the integration of the multiplet occurring in the range of 1.39-1.61 ppm was corrected, which corresponds to 10 protons of the cyclohexyl ring.
- Description of the spectra of compounds EB170, EB171 and EB173 has been supplemented with information about the proton signals (CH, CH2) of the basic lupane system.